# Fixed points of multivalued convex contractions with application

**Abdul Rahim Khan** [1]*, **Hamed H. Al-Sulami**[2], **Muhammad Rashid**[1], **Faiza Shabbir**[1]

1 Department of Mathematics and Statistics, University of Southern Punjab, Multan, Pakistan,
2 Department of Mathematics, King Abdulaziz University, Jeddah, Saudi Arabia

* abdulrahimkhan@isp.edu.pk

**Data availability statement:** All relevant data are within the paper and its Supporting Information files.

**Funding:** The author(s) received no specific funding for this work.

## Abstract

In this work, we establish fixed point outcomes for single- valued convex contraction type mappings in the context of a b-metric space. Some of the new results are extended for a multivalued convex contraction and an F-convex contraction. Thereby, an analogue of the famous Nadler's fixed point theorem for a multivalued convex contraction mapping is obtained. The relation among various contractions is presented in a diagram for an insight in this area of investigations. We apply a special case of Theorem 2.11, to solve a nonlinear Fredholm integral equation for a Chatterjea convex contraction.

## 1 Preliminaries and introduction

The exploration of fixed point theory represents a fundamental and highly impactful area in modern mathematics. The main theme in this subject is the celebrated principle of Banach contraction, which asserts that there is a unique fixed point for each contraction mapping on a complete metric space. The proof of this principle hinges on the iterates of the contraction and the principle itself has applications in a wide range of fields like partial differential equations, integral equations, image processing, optimization and artificial intelligence. In view of paramount importance of this principle, Browder and Petryshyn [7], Istratescu [11] and Berinde [2] have introduced and studied new classes of mappings enjoying higher powers of the mapping; in particular, Istratescu has coined the term "convex contraction".

We now set out to develop results for convex contraction type single-valued as well as multi-valued mappings in the context of b-metric spaces.

### 1.1 Single-valued mappings

**Definition 1.1:** Consider a nonempty set $G$ and a mapping $S : G \to G$. A point $g \in G$ is referred to as a fixed point of $S$ if it satisfies $S(g) = g$. The set consisting of all fixed points of the mapping $S$ is represented by Fix($S$).

**Definition 1.2:** Let $(G, d)$ be a complete metric space. A map $S : G \to G$ is known as Istratescu convex contraction [11] if

$$d(S^2(g), S^2(h)) \le ad(S(g), S(h)) + bd(g, h) \quad for \ any \quad g, h \in G$$

where $a$ and $b$ are constants which satisfy $0 < a, b < 1$ and $a + b < 1$.

**Competing interests:** The authors have declared that no competing interests exist.

**Remark 1.3:** ([17],Example 2.1) (i) If $b = 0$, the convex contraction condition reduces to the well-known Banach contraction condition:

$$d(S(g), S(h)) \leq ad(g, h) \quad for \ all \quad g, h \in G$$

subject to a change of notation.

(ii) If $a = 0$, the convex contraction condition reduces to the classial "asymptotic" contraction:

$$d(S^2(g), S^2(h)) \leq bd(g, h),$$

which confirms the presence of a fixed point, even when the number 2 is replaced with any arbitrary integer $n$.

**Example 1.4:** [17] Let $G = [0, 1]$ be equipped with the usual metric of $\mathbb{R}$. Define $S : [0, 1] \to [0, 1]$ by

$$S(g) = \frac{g^2 + 1}{2}, \quad g \in [0, 1].$$

Then $S$ is not a Banach contraction, and $Fix(S) = \{1\}$. But $S$ is a convex contraction, because for $g, h \in G$, we have

$$|S^2 g - S^2 h| \leq \frac{1}{2}|Sg - Sh| + \frac{1}{4}|g - h|$$

with $a = \frac{1}{2}$ and $b = \frac{1}{4}$.

**Definition 1.5:** [16] A mapping $S : G \to G$ is defined as a generalized convex contraction if there exist a function $\alpha : G \times G \to [0, \infty)$ and constants $a, b \in [0, 1)$ satisfying $a + b < 1$, so that the subsequent condition is satisfied:

$$\alpha(g, h)d(S^2(g), S^2(h)) \leq ad(S(g), S(h)) + bd(g, h) \quad for \ every \quad g, h \in G. \tag{1}$$

If $a = 0$ and $\alpha(g, h) = 1$ in Definition 1.5, then it becomes asymptotic contraction condition of Remark 1.3.

**Definition 1.6:** [9] A continuous map $S : G \to G$ on a complete metric space $(G, d)$ is referred to as a two-sided convex contraction if there are constants $a_1, a_2, b_1, b_2 \in (0, 1)$ and the following inequality is satisfied:

$$d(S^2(g), S^2(h))$$
$$\leq a_1 d(g, S(g)) + a_2 d(S(g), S^2(g) + b_1 d(h, S(h) + b_2 d(S(h), S^2(h))$$

for all $g, h \in G$ and $a_1 + a_2 + b_1 + b_2 < 1$.

**Definition 1.7.** [10] Let $S$ be a mapping from a metric space $G$ into itself. The set

$$O(S, g) = \{S^n g : n = 0, 1, 2, \dots\}$$

is referred to as the orbit of $S$ starting at $g$. We say that $S$ is *orbitally continuous* at a point $z \in G$ if for every sequence $\{g_n\} \subset O(S, g)$, with $g \in G$, the condition

$$\lim_{n \to \infty} g_n = z \quad \implies \quad \lim_{n \to \infty} S(g_n) = S(z).$$

It is important to note that every continuous self-map on a metric space is orbitally continuous, the converse does not necessarily hold [6].

**Definition 1.8:** [9] Let $(G, d)$ denote a complete metric space. A continuous mapping $S : G \to G$ is termed a Chatterjea two-sided convex contraction if there are constants $a_1, a_2, b_1, b_2 \in (0, 1)$ and the subsequent inequality is satisfied:

$$d(S^2(g), S^2(h))$$
$$\leq a_1 d(g, S(h)) + a_2 d(S(h), S^2(h)) + b_1 d(h, S(g)) + b_2 d(S(g), S^2(g))$$

for any $g, h \in G$ and $a_1 + a_2 + b_1 + b_2 < 1$.

We now provide an example of a Chatterjea two-sided convex contraction.

**Example 1.9:** Let $G = [0, 1]$ with the metric $d(g, h) = |g - h|$. Define $S : [0, 1] \to [0, 1]$ by:

$$S(g) = \frac{3g + 1}{4}, \quad g \in [0, 1]$$

Let us calculate $S^2(g)$:

$$S^2(g) = S(S(g)) = S\left(\frac{3g + 1}{4}\right) = \frac{3\left(\frac{3g+1}{4}\right) + 1}{4} = \frac{9g + 7}{16}$$

$$d(g, S(h)) = \left|g - \frac{3h + 1}{4}\right|$$

$$d(S(h), S^2(h)) = \left|\frac{3h + 1}{4} - \frac{9h + 7}{16}\right| = \frac{3|h - 1|}{16}$$

$$d(h, S(g)) = \left|h - \frac{3g + 1}{4}\right|$$

$$d(S(g), S^2(g)) = \left|\frac{3g + 1}{4} - \frac{9g + 7}{16}\right| = \frac{3|g - 1|}{16}$$

$$d(S^2(g), S^2(h)) = \left|\frac{9g + 7}{16} - \frac{9h + 7}{16}\right| = \frac{9|g - h|}{16}.$$

$$\frac{9|g - h|}{16} \leq \frac{1}{4}\left|g - \frac{3h + 1}{4}\right| + \frac{1}{8} \cdot \frac{3|h - 1|}{16} + \frac{1}{4}\left|h - \frac{3g + 1}{4}\right| + \frac{1}{8} \cdot \frac{3|g - 1|}{16}$$

where

$$a_1 = \frac{1}{4}, \quad a_2 = \frac{1}{8}, \quad b_1 = \frac{1}{4}, \quad b_2 = \frac{1}{8}$$

Since the sum of the coefficients is:

$$a_1 + a_2 + b_1 + b_2 = \frac{1}{4} + \frac{1}{8} + \frac{1}{4} + \frac{1}{8} = \frac{3}{4} < 1,$$

therefore, $S$ is a Chatterjea two-sided convex contraction.

**Definition 1.10:** [9] A mapping $S\colon G \to G$ (complete metric space) is called Hardy and Rogers convex contraction if there exist positive integers $a_1, a_2, b_1, b_2, c_1, c_2, e_1, e_2, f_1, f_2 \in (0,1)$ and the subsequent inequality is satisfied:

$$d(S^2(g), S^2(h))$$
$$\leq a_1 d(g,h) + a_2 d(S(g), S(h)) + b_1 d(g, S(g)) + b_2 d(S(g), S^2(g)) +$$
$$c_1 d(h, S(h)) + c_2 d(S(h), S^2(h)) + e_1 d(g, S(h) + e_2 d(S(h), S^2(h))$$
$$+ f_1 d(h, S(g)) + f_2 d(S(g), S^2(g))$$

for any $g, h \in G$ and $a_1 + a_2 + b_1 + b_2 + c_1 + c_2 + e_1 + e_2 + f_1 + f_2 < 1$.

It is remarked that some of the above mentioned classes of convex contraction type mappings are independent of each other [9,11]. In case, the constants are allowed to be zero in Definition 1.10, then it reduces to Istratescu convex contraction (Definition 1.2).

**Definition 1.11:** [16] Let $S\colon G \to G$ (metric space). For any given $\epsilon > 0$, a point $g_0 \in G$ is called an *approximate fixed point* of $S$ if it fulfills the condition:

$$d(g_0, Sg_0) < \epsilon.$$

**Definition 1.12:** [7] A map $S$ defined on a metric space $(G, d)$ is termed asymptotically regular at any point $g \in G$ if

$$d(S^n(g), S^{n+1}(g)) \to 0 \text{ as } n \to \infty,$$

where $S^n(g)$ denotes the $n$-th iterate of $S$ at $g$.

**Lemma 1.13:** ([16],Lemma 2.1) Suppose that $(G, d)$ is a metric space and $S$ is an asymptotically regular mapping on $G$. Then $S$ has an approximate fixed point.

**Definition 1.14:** [14] Consider the mapping $F\colon \mathbb{R}^+ \to \mathbb{R}$ that adheres to the subsequent properties:

(a) The function $F$ is strictly increasing.
(b) A sequence $\{\alpha_n\} \subset \mathbb{R}^+$ of positive real numbers satisfies $\lim_{n\to\infty} \alpha_n = 0$ if and only if $\lim_{n\to\infty} F(\alpha_n) = -\infty$.
(c) If there exists a constant $k \in (0,1)$, then $\lim_{n\to\infty} (\alpha_n^k) F(\alpha_n) = 0$.

The class of functions $F$ that satisfies conditions (a)-(c) is represented by $\mathcal{F}$.

**Definition 1.15:** An $F$-contraction is a self-map $S$ defined over a metric space $G$ if there is a function $F \in \mathcal{F}$ and a constant $\tau > 0$ such that

$$\tau + F(d(Sg, Sh)) \leq F((g,h)),$$

for all $g, h \in G$ with $d(Sg, Sh) > 0$.

**Definition 1.16:** [26] Let $F$ be a mapping that meets requirements (a)–(c). A funtion $S\colon G \to G$ is known as an $F$-Kannan mapping if the following hold:

(K1) $Sg \neq Sh \implies Sg \neq g$ or $Sh \neq h$

(K2) $\exists \gamma > 0$ such that

$$\gamma + F(d(Sg, Sh)) \leq F\left(\frac{d(g, Sg) + d(h, Sh)}{2}\right)$$

for all $g, h \in G$, with $Sg \neq Sh$.

**Example 1.17:** ([26],Lemma 12) Consider a metric space $(G, d)$ and F-Kannan mapping $S : G \to G$. Then,

$$d(S^n(g), S^{n+1}(g)) \to 0 \quad \text{as} \quad n \to \infty \quad \text{for all} \quad g \in G.$$

So an F-Kannan mapping is asymptotically regular.

Here are some well known results for convex contraction type mappings.

**Theorem 1.18:** ([11], Theorem 1.2) Every convex contraction mapping defined on a complete metric space has a unique fixed point.

**Theorem 1.19:** ([5], Theorem 2.1) If $S$ is a self-map on a complete metric space $(G, d)$, $g, h \in G$, $a_1, a_2, b_1, b_2, c_1, c_2 \in (0, 1)$ and $0 \leq a_1 + a_2 + b_1 + b_2 + c_1 + c_2 < 1$, then we have

$$d(S^2(g), S^2(h))$$
$$\leq a_1 d(g, h) + a_2 d(S(g)), S(h)) + b_1 d(g, S(g)) + b_2 d(S(g), S^2(g))$$
$$+ c_1 d(h, S(h)) + c_2 d(S(h), S^2(h)).$$

Suppose that $S$ is $k$-continuous for $k \in \mathbb{N}$ [12]. Then $S$ admits a unique fixed point.

**Theorem 1.20:** ([6],Theorem 2.1) Let $S$ be a self-map of a complete metric space $(G, d)$ such that for each $g, h \in G$;

$$d(S^m(g), S^m(h)) \leq a_j d(S^j(g), S^j(h)),$$

where the constants $a_0, a_1, \ldots, a_{m-1}$ are non-negative and their sum satisfies $\sum_{j=0}^{m-1} a_j < 1$. The map $S$ has a unique fixed point if it is either orbitally continuous or k-continuous.

**Theorem 1.21:** ([9], Theorem 2.5) Let $(G, d)$ be a complete metric space and $S$ be a self-map on $G$ satisfying the condition:

$$d(S^2(g), S^2(h))$$
$$\leq a_1 d(g, S(h)) + a_2 d(S(h), S^2(h)) + b_1 d(h, S(g)) + b_2(S(g), S^2(g)),$$

for all $g, h \in G$, where $a_1, a_2, b_1, b_2 \in (0, 1)$ and $a_1 + a_2 + b_1 + b_2 < 1$. If $S$ is orbitally continuous, then $S$ admits a unique fixed point in $G$. Moreover, for any initial point $g_0 \in G$, the Picard iterations sequence $\{g_n\}_{n=0}^{\infty}$, defined by $g_{n+1} = S(g_n)$ for $n \geq 0$, converges to a fixed point of $S$ in $G$.

## 1.2 Multivalued mappings

**Definition 1.22:** Let $(G, d)$ be a metric space and $CB(G)$ represent the family of closed and bounded subsets of $G$. A mapping $S : G \to CB(G)$ has $g_0$ as it's fixed point if $g_0 \in Sg_0$.

**Definition 1.23** [12] A map $S$ on a metric space $(G, d)$ is known as asymptotically regular at a point $g \in G$ if

$$H(S^n(g), S^{n+1}(g)) \to 0 \quad as \quad n \to \infty,$$

$H$ is the Hausdorff metric given by
$H(A, B) = max\{sup\{d(a, B) : a \in A\}, sup\{d(b, A) : b \in B\}\}$
where $d(a, B) = inf\{d(a, b) : b \in B\}$.

**Definition 1.24:** Consider the metric space $(G, d)$. A mapping $S : G \to CB(G)$ is called convex contraction if

$$H(S^2(g), S^2(h)) \leq ad(S(g), S(h)) + bd(g, h)$$

for all $g, h \in G$, where $a, b$ are the constants that fulfill $0 < a, b < 1$ and $a + b < 1$.

**Definition 1.25:** Let $(G, d)$ be a metric space. A mapping $S : G \to CB(G)$ is called Chatterjea two-sided convex contraction if the following holds:

$$H(S^2(g), S^2(h))$$
$$\leq a_1 d(g, S(h)) + a_2 d(S(h), S^2(h)) + b_1 d(h, S(g)) + b_2 d(S(g), S^2(g)),$$

for every $g, h \in G$, $a_1, a_2, b_1, b_2 \in (0, 1)$ and $a_1 + a_2 + b_1 + b_2 < 1$.

**Definition 1.26:** Let $(G, d)$ be a complete metric space and $S : G \to CB(G)$ be a mapping. The mapping $S$ is said to be a weak convex contraction if it satisfies:

$$H(S^2(g), S^2(h))$$
$$\leq a_0 d(g, h) + b_0 d(h, S(g)) + a_1 d(S(g), (Sh)) + b_1 d(S(h), S^2(g))$$

for all $g, h \in G$, $b_0, b_1 \geq 0$, $0 < a_0, a_1 < 1$ and $a_0 + a_1 < 1$.

**Definition 1.27:** [2] Let $(G, d)$ be a metric space and $S : G \to CB(G)$ be a multivalued map. If $\theta \in (0, 1)$ and $L \geq 0$, then $S$ is a multivalued weak contraction if

$$H(Sg, Sh) \leq \theta d(g, h) + Ld(h, Sg),$$

holds for each pair of points $g, h \in G$.

For ease of reference, we provide proof of the result to follow.

**Lemma 1.28:** ([2], Lemma 1) Consider a metric space $(G, d)$, two subsets $A, B \subseteq G$ and a fixed constant $q > 1$. Then, for each $a \in A$, there exists an element $b \in B$ such that

$$d(a, b) \leq qH(A, B).$$

**Proof:** For $H(A, B) = 0$, the result holds for $b = a$ with $a \in B$.
Set $\epsilon = (k^{-1} - 1)H(A, B) > 0$ where $k < 1$.
According to the definitions of $d(a, B)$ and $H(A, B)$, for all $\epsilon > 0$, there exists an element $b \in B$ such that

$$d(a, b) \leq d(a, B) + \epsilon \leq H(A, B) + \epsilon. \tag{*}$$

Putting the selected value of $\epsilon$ in the above inequality, we get the result with $q = k^{-1}$.

Fixed point results for single-valued asymptotically regular and convex contraction type mappings have been considered by Berinde and Pacurar [3], Bisht and Hussain [4] and Khan and Oyetunbi [12] while Khan and Oyetunbi [13], Karakaya and Sekman [15] and Sgroi and Vetro [23] have studied these results for multivalued mappings.In this paper, we will extend Theorem 1.21 for a Chatterjea two-sided convex contraction in a b-metric space. On the one hand, we apply our new result to solve a non-linear Fredholm integral equation and on the other hand, we find it's multivalued version. A fixed point result for generalized convex contraction on a b-metric space is proved in Theorem 2.5. We established a multivalued version of Theorem 1.18, the fundamental result of Istratescu for a convex contraction.

Recently, Nallaselli et al [19], Özkan et al [21] and Ricinschi et al [22] have studied convex contractions on metric spaces and b-metric spaces. It is remarked that our results and techniques are different from their ones and are more focused on the development of fixed point results for a multivalued convex contraction.

## 2 Fixed points results

### 2.1 Single-valued mappings

The concept of a b-metric space was presented by Czerwik [8] as follows:

**Definition 2.1:** Let $G$ be a nonempty set and $s \geq 1$ be a fixed real number. A function $d:$ $G \times G \to \mathbb{R}^+$ is said to be a b-metric if, for all $g, h, z \in G$, the subsequent conditions hold:

(b$_1$)  $d(g, h) = 0, \iff g = h$
(b$_2$)  $d(g, h) = d(h, g)$
(b$_3$)  $d(g, h) \leq s[d(g, z) + d(z, h)]$.

The triplet $(G, d, s)$ is then referred to as a b-metric space.

Note that when $s = 1$, a b-metric space becomes a metric space. But, in general, the converse does not hold.

We extend Theorem 2.5 in [9] for b-metric spaces as follows.

**Theorem 2.2:** Let $(G, d, s)$ be a complete b-metric space with coefficient $s \geq 1$ and $S$ be a self-map on $G$ satisfying the condition;

$$d(S^2(g), S^2(h))$$
$$\leq a_1 d(g, S(h)) + a_2 d(S(h), S^2(h)) + b_1 d(h, S(g)) + b_2 d(S(g), S^2(g)), \tag{2}$$

for all $g, h \in G$, $a_1, a_2, b_1, b_2 \in (0, 1)$, $a_1 + a_2 + b_1 + b_2 < 1$ and $2sa_1 + b_2 + a_2 < 1$.

If $S$ is orbitally continuous, then it admits a unique fixed point in $G$. For any initial point $g_0 \in G$, the sequence $\{g_n\}_{n=0}^{\infty}$, defined recursively by $g_{n+1} = Sg_n$ for $n \geq 0$, converges to the unique fixed point of $S$ in $G$.

**Proof:** Let $g_0 \in G$ be an arbitrary point. Define a sequence $\{g_n\}$ by

$$g_1 = S(g_0),$$
$$g_2 = S(g_1),$$
$$g_3 = S(g_2),$$
$$\dots$$
$$g_{n+1} = S(g_n) = S^n(g_0),$$

for all $n = 0, 1, 2, \dots$.

Put $g = g_0$, $h = S(g_0)$ in (2), we get

$$d(S^2(g_0), S^3(g_0))$$
$$\leq a_1 d(g_0, S^2(g_0)) + a_2 d(S^2(g_0), S^3(g_0)) + b_1 d(S(g_0), S(g_0)) +$$
$$b_2 d(S(g_0), S^2(g_0))$$
$$\leq a_1 d(g_0, S^2(g_0)) + a_2 d(S^2(g_0), S^3(g_0)) + b_2 d(S(g_0), S^2(g_0))$$
$$\leq a_1 s[d(g_0, S(g_0)) + d(S(g_0), S^2(g_0))] + a_2 d(S^2(g_0), S^3(g_0)) +$$
$$b_2 d(S(g_0), S^2(g_0))$$
$$\leq s a_1 d(g_0, S(g_0)) + s a_1 d(S(g_0), S^2(g_0)) + a_2 d(S^2(g_0), S^3(g_0)) +$$
$$b_2 d(S(g_0), S^2(g_0))$$
$$\leq s a_1 d(g_0, S(g_0)) + (s a_1 + b_2) d(S(g_0), S^2(g_0)) + a_2 d(S^2(g_0), S^3(g_0))$$
$$\leq (2 s a_1 + b_2) \max\{d(x_0, S(g_0)), d(S(g_0), S^2(g_0))\} + a_2 d(S^2(g_0), S^3(g_0)).$$
$$\leq \frac{(2 s a_1 + b_2)}{1 - a_2} \max\{d(g_0, S(g_0)), d(S(g_0), S^2(g_0))\},$$
$$\leq \frac{2 s a_1 + b_2}{1 - a_2} k \quad where \quad k = \max\{d(g_0, S(g_0)), d(S(g_0), S^2(g_0))\}.$$

Now substitute $g = S(g_0)$ and $h = S^2(g_0)$ in (2), and get:

$$d(S^3(g_0), S^4(g_0))$$
$$\leq a_1 d(S(g_0), S^3(g_0)) + a_2 d(S^3(g_0), S^4(g_0)) + b_1 d(S^2(g_0), S^2(g_0)) +$$
$$b_2 d(S^2(g_0), S^3(g_0))$$
$$\leq a_1 d(S(g_0), S^3(g_0)) + a_2 d(S^3(g_0), S^4(g_0)) + b_2 d(S^2(g_0), S^3(g_0))$$
$$\leq s a_1 [d(S(g_0), S^2(g_0)) + d(S^2(g_0), S^3(g_0))] + a_2 d(S^3(g_0), S^4(g_0)) +$$
$$b_2 d(S^2(g_0), S^3(g_0))$$
$$\leq s a_1 d(S(g_0), S^2(g_0)) + s a_1 d(S^2(g_0), S^3(g_0)) + a_2 d(S^3(g_0), S^4(g_0)) +$$
$$b_2 d(S^2(g_0), S^3(g_0))$$
$$\leq \frac{2 s a_1 + b_2}{1 - a_2} k.$$

By continuing this process, we obtain:

$$d(S^n(g_0), S^{n+1}(g_0)) \leq \left(\frac{2 s a_1 + b_2}{1 - a_2}\right)^{n-2} k$$
$$\implies d(S^n(g_0), S^{n+1}(g_0)) \leq \gamma^{n-2} k, \quad where \quad \gamma = \frac{2 s a_1 + b_2}{1 - a_2} < 1.$$

We now demonstrate that $\{g_n\}$ forms a Cauchy sequence in $G$. For $m < n$, we get

$$d(S^n(g_0), S^m(g_0))$$
$$\leq s[d(S^n(g_0), S^{n+1}(g_0)) + d(S^{n+1}(g_0), S^m(g_0))]$$
$$\leq s d(S^n(g_0), S^{n+1}(g_0)) + s d(S^{n+1}(g_0), S^m(g_0))$$
$$\leq s d(S^n(g_0), S^{n+1}(g_0)) + s^2 d(S^{n+1}(g_0), S^{n+2}(g_0)) +$$
$$s^3 d(S^{n+2}(g_0), S^{n+3}(g_0)) + ... + s^{(m-1)} d(S^{m-1}(g_0), S^m(g_0))$$

$$\le s\gamma^{(n-2)}k + s^2\gamma^{(n-1)}k + s^3\gamma^n k + \dots$$

$$\le s\gamma^{(n-2)}\left[1 + s\gamma + s^2\gamma^2 + \dots\right]k$$

$$\le \frac{s\gamma^{(n-2)}}{1 - s\gamma}k.$$

$$\implies d(S^n(g_0), S^m(g_0)) \le \frac{s\gamma^{(n-2)}}{1 - s\gamma}k$$

As $n \to \infty$, in the view of $s\gamma < 1$, $d(S^n(g_0), S^m(g_0)) \to 0$.

Therefore $d(S^n(g_0), S^{n+1}(g_0))$ is a Cauchy sequence. Since $G$ is complete, there exists a point $g \in G$ such that $g_n \to g$ as $n \to \infty$. So applying orbital continuity of $S$, we get

$$\lim_{n \to \infty} S^{(n+1)}(g_0) = S(g).$$

This shows that $g$ is a fixed point of $S$.

For it's uniqueness, assume on the contrary that $u$ is another fixed point of $S$ such that $g \ne u$.

$$
\begin{aligned}
d(g, u) = d(S^2(g), S^2(u)) &\le a_1 d(g, S(u) + a_2 d(S(u), S^2(u)) + b_1 d(u, S(g)) + \\
&\quad b_2 d(S(g), S^2(g)) \\
&\le a_1 d(g, S(u) + a_2 d(S(u), S(u)) + b_1 d(u, S(g)) + b_2 d(S(g), S(g)) \\
&\le a_1 d(g, S(u) + b_1 d(u, S(g)) \\
&\le a_1 d(g, u) + b_1 d(u, g) \\
&\le (a_1 + b_1) d(g, u). \\
&\implies d(g, u) - (a_1 + b_1) d(g, u) \le 0 \\
&(1 - a_1 - b_1) d(g, u) \le 0
\end{aligned}
$$

As $1 - a_1 - b_1 < 0$, so d(g,u)=0 implies $g = u$ which is a contradiction. Hence $S$ has a unique fixed point in $G$.

An extension of Corollary 1 in [1] for b-metric spaces is presented below.

**Corollary 2.3:** Let $S$ be a self-map on a complete b-metric space $(G, d, s)$ satisfying (2) in Theorem 2.2 with $a_1 + a_2 + b_1 + b_2 < 1$. Then both $S$ and $S^2$ have a unique common fixed point (i.e. $S(z) = S^2(z) = z$).

**Proof:** By Theorem 2.2, there exists a unique fixed point $z$ of $S$ and so $z \in Fix(S)$.

Now, we aim to show that $S$ and $S^2$ have a unique common fixed point. Clearly, $F(S) \subseteq F(S^2)$. Suppose $u \in F(S^2)$ and $u \ne z$. Then $S^2(u) = u$.

Assume, for a contradiction, that $S(u) \ne u$. Since $S^2(u) = u$, therefore by (2), applied to the pair $u$ and $S(u)$, we get

$$
\begin{aligned}
d(S(u), u) &= d(S^2(S(u)), S^2(u)), \\
d(S(u), u) &\le a_1 d(u, S(u)) + a_2 d(S(u), S^2(u)) + b_1 d(S(u), u) + b_2 d(u, S(u)). \\
d(S(u), u) &\le a_1 d(u, S(u)) + a_2 d(S(u), u) + b_1 d(S(u), u) + b_2 d(u, S(u)), \\
d(S(u), u) &\le (a_1 + b_2) d(u, S(u)) + (a_2 + b_1) d(S(u), u).
\end{aligned}
$$

Now, if $d(S(u), u) \ne 0$, the above inequality implies that $d(S(u), u)$ is smaller than itself, which is a contradiction. Thus, $S(u) = u$, and hence $u \in Fix(S)$. Therefore, $F(S) = F(S^2)$, and

since $z$ is the only fixed point of $S$, it must also be the only fixed point of $S^2$. This shows that both $S$ and $S^2$ possess a unique common fixed point.

**Definition 2.4** [16] Consider a self-map $S$ defined on a nonempty set $G$, along with a function $\alpha : G \times G \to [0, \infty)$. $S$ is $\alpha$-admissible if for $g, h \in G$,

$$\alpha(g, h) \geq 1 \implies \alpha(Sg, Sh) \geq 1.$$

We now extend ([16],Theorem 3.1) in the framework of b-metric spaces.

**Theorem 2.5:** Consider a $b$-metric space $(G, d, s)$ with $s \geq 1$, $S$ a generalized convex contraction on $G$ characterized by the base mapping $\alpha$. Assume that $S$ satisfies the $\alpha$-admissibility condition and that there exists a point $g_0 \in G$ such that $\alpha(g_0, S(g_0)) \geq 1$. Under these assumptions, $S$ possesses an approximate fixed point. Furthermore, if $S$ is continuous and $(G, d, s)$ is complete, then $S$ has a fixed point.

**Proof:** Let $g_0 \in G$ be such that $\alpha(g_0, S(g_0)) \geq 1$. Define $\{g_n\}$ in $G$ as before to get

$$g_{n+1} = S(g_n) = S^n(g_0) \quad \text{for all } n \geq 0.$$

If $g_n = g_{n+1}$ for some $n$, the proof is complete. Otherwise, assume $g_n \neq g_{n+1}$. Given that $S$ is $\alpha$-admissible, we have the condition $\alpha(g_0, S(g_0)) \geq 1$ for all $n$. Let $v = d(S(g_0), S^2(g_0)) + d(g_0, S(g_0))$ and $\lambda = a + b$.

Then $d(S(g_0), S^2(g_0)) \leq v$.

Now put $g = g_0$, $h = S(g_0)$ in (1) and get

$$\begin{aligned}
d(S^2(g_0), S^3(g_0)) &\leq \alpha(S(g_0), g_0)\, d(S^2(g_0), S^3(g_0)) \\
&\leq a\, d(S(g_0), S^2(g_0)) + b\, d(g_0, S(g_0)) \\
&\leq \lambda v
\end{aligned}$$

Again put $g = Sg_0$ and $h = S^2 g_0$ in (1) to obtain.

$$\begin{aligned}
d(S^3(g_0), S^4(g_0)) &\leq \alpha(S(g_0), S^2(g_0)) d(S^3(g_0), S^4(g_0)) \\
&\leq a d(S^2(g_0), S^3(g_0)) + b d(S(g_0), S^2(g_0)) \\
&\leq \lambda v
\end{aligned}$$

Similarly,

$$d(S^4(g_0), S^5(g_0)) \leq \lambda^2 v$$

And also,

$$d(S^5(g_0), S^6(g_0)) \leq \lambda^2 v$$

By continuing this procedure, we get

$$d(S^m(g_0), S^{m+1}(g_0)) \leq \lambda^l v$$

where $m = 2l$ or $m = 2l + 1$. Consequently, $d(S^m g_0, S^{m+1} g_0) \to 0$ as $n \to \infty$. This implies that $S$ is asymptotically regular. By Lemma 1.13, $S$ possesses an approximate fixed point. Now, suppose that $S$ is continuous, and the space $(G, d)$ is a complete $b$-metric space.

To demonstrate that $\{g_n\}$ is a Cauchy sequence, consider $m$ and $n$ as positive integers such that $m < n$. Let $m = 2l$ and $n = 2p$, where $l \geq 1$ and $p \geq 2$.

$$
\begin{aligned}
&d(S^m(g_0), S^n(g_0)) \\
&\quad \leq s[d(S^m(g_0), S^{m+1}(g_0)) + d(S^{m+1}(g_0), S^n(g_0))] \\
&\quad \leq sd(S^m(g_0), S^{m+1}(g_0)) + sd(S^{m+1}(g_0), S^n(g_0)) \\
&\quad \leq sd(S^m(g_0), S^{m+1}(g_0)) + s[d(S^{m+1}(g_0), S^{m+2}(g_0)) + d(S^{m+2}(g_0), S^n(g_0))] \\
&\quad \leq sd(S^m(g_0), S^{m+1}(g_0)) + s^2d(S^{m+1}(g_0), S^{m+2}(g_0)) + s^2d(S^{m+2}(g_0), S^n(g_0)) \\
&\quad \leq sd(S^m(g_0), S^{m+1}(g_0)) + s^2d(S^{m+1}(g_0), S^{m+2}(g_0)) + ... + s^{n-1} \\
&d(S^{n-1}(g_0), S^n(g_0)) \\
&\quad = sd(S^{2l}(g_0), S^{2l+1}(g_0)) + s^2d(S^{2l+1}(g_0), S^{2l+2}(g_0)) + ... + s^{2p-1} \\
&d(S^{2p-1}(g_0), S^{2p}(g_0)) \\
&\quad \leq s\lambda^l v + s^2\lambda^l v + s^3\lambda^{l+1} v + ... + s^{2p-1}\lambda^{p-1} v \\
&\quad \leq (s\lambda^l v + s^3\lambda^{l+1} v + s^5\lambda^{l+2} + \cdots) + (s^2\lambda^l v + s^4\lambda^{l+1} v + s^6\lambda^{l+2} + \cdots) \\
&\quad \leq s\lambda^l(1 + s^2\lambda + \cdots)v + s^2\lambda^l(1 + s^2\lambda^l + \cdots)v \\
&\quad \leq (s + s^2)\lambda^l(1 + s^2\lambda + \cdots)v \\
&\quad \leq (s + s^2)\lambda^l \frac{1}{1 - s^2\lambda}v
\end{aligned}
$$

For $m = 2l + 1$ and $n = 2p$, where $p \geq 1$ and $l \geq 1$, with the condition that $m < n$, we obtain:

$$
\begin{aligned}
&d(S^m(g_0), S^n(g_0)) \\
&\quad \leq sd(S^m(g_0), S^{m+1}(g_0)) + s^2d(S^{m+1}(g_0), S^{m+2}(g_0)) + ... + s^{n-1} \\
&d(S^{n-1}(g_0), S^n g_0) \\
&\quad \leq sd(S^{2l+1}(g_0), S^{2l+2}(g_0)) + s^2d(S^{2l+2}(g_0), S^{2l+3}(g_0)) + ... + s^{2p-1} \\
&d(S^{2p-1}(g_0), S^{2p}(g_0)) \\
&\quad \leq s\lambda^l v + s^2\lambda^l v + s^3\lambda^{l+1} v + ... \\
&\quad \leq (s\lambda^l v + s^3\lambda^{l+1} v + s^5\lambda^{l+2} ...) + (s^2\lambda^l v + s^4\lambda^{l+1} v + s^6\lambda^{l+2} ...) \\
&\quad \leq s\lambda^l(1 + s^2\lambda + ...)v + s^2\lambda^l(1 + s^2\lambda^l + ...)v \\
&\quad \leq (s + s^2)\lambda^l\{1 + s^2\lambda + ...\}v \\
&\quad \leq (s + s^2)\lambda^l\left(\frac{1}{1 - s^2\lambda}\right)v
\end{aligned}
$$

Similarly, let $m = 2l + 1$ and $n = 2p + 1$, where $p \geq 2$ and $l \geq 1$.

$$
d(S^m(g_0), S^n(g_0)) \leq (s + s^2)\lambda^l\left(\frac{1}{1 - s^2\lambda}\right)^v
$$

For $l \to \infty$, in all the cases, we obtain $d(S^m(g_0), S^n(g_0)) \to 0$ (in view of $s^2\lambda < 1$). That is, $\{g_n\}$ is a Cauchy sequence in $G$. As $G$ is complete, so there exists $z \in G$ such that $g_n = S^n g_n \to z \in G$ as $n \to \infty$. If $S$ is continuous, then $Sg_n \to Sz$, i.e., $g_{n+1} \to Sz$. Hence $Sz = z$.

Now we need to prove that $S$ has a unique fixed point in $G$. Assume, on the contrary that $z^*$ is a fixed point of $S$ such that $z \neq z^*$.

Taking $z = g$ and $z^* = h$ in (1), we get by hypothesis $\alpha(z, z^*) \geq 1$,

$$
\begin{aligned}
d(z, z^*) = d(S^2 z, S^2 z^*) & \\
& \leq \alpha(z, z^*) d(S^2 z, S^2 z^*) \\
& \leq a d(Sz, Sz^*) + b d(z, z^*) \\
& \leq a d(z, z^*) + b d(z, z^*) \\
& \leq (a + b) d(z, z^*),
\end{aligned}
$$

$$(1 - a - b) d(z, z^*) \leq 0.$$

In view of $(1 - a - b) < 0$, $d(z, z^*) = 0$. Hence $z = z^*$, which is a contradiction. Hence $S$ has a unique fixed point in $G$.

**Lemma 2.6:** ([18], Lemma 4) Let $k \in \mathbb{N}$ and $\{g_n\}$ represent a sequence in a $b$-metric space $(G, d, s)$ such that

$$d(g_{n+k}, g_{n+k-1}) \leq \sum_{i=0}^{k-1} a_i d(g_{n+i}, g_{n+i-1}) \tag{3}$$

for all $n \in \mathbb{N}$, where $a_i \geq 0$ and

$$\sum_{i=0}^{k-1} a_i < 1.$$

Then $\{g_n\}$ is Cauchy.

The following result improves ([18], Theorem 1).

**Theorem 2.7:** Let $S : G \to G$ be a convex contraction of order $k$ defined on a complete $b$-metric space $(G, d, s)$, such that

$$d(S^k g, S^k h) \leq \sum_{i=0}^{k-1} a_i d(S^i g, S^i h)$$

for all $g, h \in G$, where $a_i$ is non-negative and the sum satisfies $\sum_{i=0}^{k-1} a_i < 1$. If $S$ exhibits orbital continuity, then $S$ admits a unique fixed point in $G$.

**Proof** Let $g_0 \in G$ be arbitrary. Define a sequence $\{g_n\}$, as before, to get

$$g_{n+1} = S(g_n).$$

Now,

$$
\begin{aligned}
d(g_{n+k}, g_{n+k-1}) = d(S^k(g_n), S^k(g_{n-1})) & \\
& \leq \sum_{i=0}^{k-1} a_i d(S^i(g_n), S^i(g_{n-1})) \\
& \leq \sum_{i=0}^{k-1} a_i d(S^i(g_{n+i}), S^i(g_{n+i-1}))
\end{aligned}
$$

for every $n \in \mathbb{N}$, where $a_i$ is non-negative, and the sum $\sum_{i=0}^{k-1} a_i < 1$.

By Lemma 2.6, the sequence $\{g_n\}$ forms a Cauchy sequence in the complete space $G$. Hence, there exists a point $z \in G$ such that $g_n \to z$ as $n \to \infty$. Furthermore, since $S(g_n) \to z$, the orbital continuity of $S$ ensures that $\lim_{n \to \infty} S(g_n) = S(z)$. Consequently, we have $S(z) = z$, implying that $z$ is a fixed point of $S$, and uniqueness of $z$ follows as before.

## 2.2 Multi-valued mappings

**Lemma 2.8:** [25] Let $(G, d, s)$ be a $b$-metric space and $\mathrm{CB}(G)$ denotes the class of all nonempty, closed, and bounded subsets of $G$. For any $U, V \in \mathrm{CB}(G)$, the following are satisfied:

1.

$$d(a, U) \leq H(U, V), \quad a \in U;$$

2.

$$\text{For } \varepsilon > 0 \text{ and } a \in U, \exists \quad b \in V \text{ such that } d(a, b) \leq H(U, V) + \varepsilon.$$

**Definition 2.9** (cf. [25],Definition 2.4) Let $G$ be an arbitrary nonempty set and $s \geq 1$ be a fixed real number. A strong $b$-metric on $G$ is a function $d : G \times G \to \mathbb{R}$, satisfying the following axioms for $g, h, z \in G$ :

(a) $d(g, h) \geq 0$;
(b) $d(g, h) = 0 \Leftrightarrow g = h$;
(c) $d(g, h) = d(h, g)$;
(d) $d(g, h) \leq d(g, z) + sd(z, h)$.

The triplet $(G, d, s)$ denotes a strong $b$-metric space.

We establish a multivalued version of Theorem 1.18, a classical result of Istratescu [11].

**Theorem 2.10:** Let $(G, d, s)$ be a complete strong $b$-metric space and $S : G \to CB(G)$ be an asymptotically regular convex contraction. Then there exists $h \in G$ such that $h \in Sh$.

**Proof:** Let $g_0 \in G$. Then $Sg_0 \neq \phi$ is a closed and bounded subset of $G$. Furthermore, let $g_1 \in Sg_0$ and $Sg_1 \neq \phi$ be a closed and bounded subset of $G$. By Lemma 2.8 (2), there exists $g_2 \in Sg_1$ such that

$$d(g_1, g_2) \leq H(S^2(g_0), S^2(g_1)) + \alpha. \tag{4}$$

Using the definition of convex contraction and asymptotic regularity of $S$, we get

$$d(g_1, g_2) \leq \alpha d(g_0, g_1) + \beta d(S(g_0), S(g_1)) + \alpha$$
$$\leq \alpha d(g_0, g_1) + \beta d(S(g_0), S^2 g_0)) + \alpha$$
$$d(g_1, g_2) \leq \alpha d(g_0, g_1) + \alpha. \tag{5}$$

Now, $Sg_2 \neq \phi$ is a closed and bounded subset of $G$, so there exists $g_3 \in Sg_2$ such that

$$d(g_2, g_3) \leq H(S^2(g_1), S^2(g_2)) + \alpha^2.$$

As before,

$$d(g_2, g_3) \leq \alpha d(g_1, g_2) + \beta d(S(g_1), S(g_2)) + \alpha^2$$

$$\leq \alpha d(g_1, g_2) + \beta d(S(g_1), S^2(g_1)) + \alpha^2$$
$$d(g_2, g_3) \leq \alpha d(g_1, g_2) + \alpha^2. \tag{6}$$

Similarly,

$$d(g_3, g_4) \leq H(S^2(g_2), S^2(g_3)) + \alpha^3$$
$$\leq \alpha d(g_2, g_3) + \alpha^3. \tag{7}$$

Using (6), we have

$$d(g_3, g_4) \leq \alpha(\alpha d(g_1, g_2) + \alpha^2) + \alpha^3$$
$$d(g_3, g_4) \leq \alpha^2 d(g_1, g_2) + 2\alpha^3$$
$$\leq \alpha^2 d(g_1, g_2) + 2\alpha^3$$
$$d(g_3, g_4) \leq \alpha^2(\alpha d(g_0, g_1) + \alpha) + 2\alpha^3$$
$$\leq \alpha^3 d(g_0, g_1) + 3\alpha^3.$$

In general,

$$d(g_n, g_{n+1}) \leq \alpha^n d(g_0, g_1) + n\alpha^n.$$

For convenience, we set

$$d(g_n, g_{n+1}) = d_n,$$

so the above result can be written as

$$d_n \leq \alpha^n d_0 + n\alpha^n. \tag{8}$$

For $m, n \in \mathbb{N}$, $m \geq n$, we have

$$d(g_n, g_m)$$
$$\leq d(g_n, g_{n+1}) + sd(g_{n+1}, g_{n+2}) + s^2 d(g_{n+2}, g_{n+3}) + \cdots + s^{m-n-1} d(g_{m-1}, g_m).$$

Using (8), we get

$$d(g_n, g_m) \leq \alpha^n d_0 + s\alpha^{n+1} d_0 + s^2 \alpha^{n+2} d_0 + \cdots + s^{m-n-1}\alpha^{m-1} d_0$$
$$+ n\alpha^n + (n+1)\alpha^{n+1} + \cdots + (m-1)\alpha^{m-1}$$
$$\leq \alpha^n d_0 (1 + \alpha s + (\alpha s)^2 + (\alpha s)^3 + \cdots + s^{m-n-1}\alpha^{m-n-1})$$
$$+ \sum_{i=n}^{m-1} i s^{i-n} \alpha^i.$$

In the limiting case, when $m, n \to \infty$,

$$d(g_n, g_m) = 0.$$

So $\{g_n\}$ is a Cauchy sequence in $G$. By completeness of $G$, there exists $h \in G$ such that $g_n \to h$. Now we will prove that $h$ is a fixed point of $S$.

$$d(h, Sh) \leq d(h, g_n) + sd(g_n, S(h)).$$

By Lemma 2.8 (1),

$$\leq d(h, g_n) + sH(S^2(g_{n-1}), S^2(h))$$
$$\leq d(h, g_n) + s[\alpha d(g_{n-1}, h) + \beta d(S(g_{n-1}), (Sh))].$$

In the limiting case, when $n \to \infty$,

$$d(h, Sh) \leq 0,$$
$$d(h, Sh) = 0.$$

Now $Sh$ is closed and so $h \in Sh$. Hence, $h$ is a fixed point of $S$ as desired.

Here is a multivalued version of Theorem 2.2 for a convex contraction (see also [20],[24],[27]).

**Theorem 2.11:** Let $(G, d)$ be a complete metric space and let $S : G \to CB(G)$ be a multivalued $F$-convex contraction satisfying:

$$2\tau + F(H(S^2(g), S^2(h))) \leq F(a_1 d(g, S(h)) + a_2 d(S(h), S^2(h)) + b_1 d(h, S(g)) + \\ b_2 d(S(g), S^2(g))) \tag{9}$$

for all $g, h \in G$, $a_1, a_2, b_1, b_2 \in (0, 1)$ and $a_1 + a_2 + b_1 + b_2 < 1$. Then $S$ has a fixed point.

**Proof:** Let $g_0 \in G$ be an arbitrary point of $G$ and choose $g_1 \in S(g_0)$. If $g_1 \in S(g_1)$, then $g_1$ is a fixed point of $S$ and nothing to prove.

Assume that $g_1 \notin S(g_1)$. Then $S(g_0) \neq S(g_1)$.

$$2\tau + Fd(S^2(g_0), S^3(g_0)) \leq 2\tau + F(H(S^2(g_0), S^3(g_0)) + \tau$$
$$Fd(S^2(g_0), S^3(g_0)) \leq F(H(S^2(g_0), S^3(g_0))) + \tau$$

Let $g = g_0$, $h = S(g_0)$ in (9) and set $K = \max\{d(x_0, S(g_0)), d(S(g_0), S^2(g_0))\}$,

$$F(H(S^2(g_0), S^3(g_0)))$$
$$\leq F(a_1 d(g_0, S^2(g_0)) + a_2 d(S^2(g_0), S^3(g_0)) + b_1 d(S(g_0), S(g_0))$$
$$+ b_2 d(S(g_0), S^2(g_0))$$
$$\leq F(a_1 d(g_0, S^2(g_0)) + a_2 d(S^2(g_0), S^3(g_0)) + b_2 d(S(g_0), S^2(g_0))$$
$$\leq F(a_1 d(g_0, S(g_0)) + a_1 d(S(g_0), S^2(g_0)) + b_2 d(S(g_0), S^2(g_0))$$
$$+ a_2 H(S^2(g_0), S^3(g_0))$$
$$\leq F((a_1 + a_2 + b_2) \max\{d(x_0, S(g_0)), d(S(g_0), S^2(g_0))\} + a_2 H(S^2(g_0), S^3(g_0))$$
$$\leq F\left(\frac{2a_1 + b_2}{1 - a_2} K\right) \tag{10}$$

Since $F$ is strictly increasing and $\tau$ is greater than zero, (10) becomes

$$< \frac{2a_1 + b_2}{1 - a_2} K.$$

$$So \quad H(S^2(g_0), S^3(g_0)) \leq \lambda K \tag{11}$$

where $\lambda = \frac{2a_1+b_2}{1-a_2}$. Substituting $g = S(g_0)$, $h = S^2(g_0)$ in (9), we have

$$2\tau + Fd(S^3(g_0), S^4(g_0))$$
$$\leq 2\tau + F(H(S^3(g_0), S^4(g_0)) + \tau$$
$$\leq F(a_1 d(S(g_0), S^3(g_0)) + a_2 d(S^3(g_0), S^4(g_0)) + b_1 d(S^2(g_0), S^2(g_0)) +$$
$$\quad b_2 d(S^2(g_0), S^3(g_0)))$$
$$\leq F(a_1 d(S(g_0), S^2(g_0)) + a_1 d(S^2(g_0), S^3(g_0)) + b_2 d(S^2(g_0), S^3(g_0)) +$$
$$\quad a_2 H(S^3(g_0), S^4(g_0)))$$
$$\leq F\left(\frac{2a_1 + b_2}{1 - a_2} K\right)$$

Given that $F$ is strictly increasing and $\tau > 0$

$$< \frac{2a_1 + b_2}{1 - a_2} K$$
$$H(S^3(g_0), S^4(g_0)) \leq \lambda K$$

Similarly, we can show that

$$H(S^4(g_0), S^5(g_0)) \leq \lambda^2 K$$
$$H(S^5(g_0), S^6(g_0)) \leq \lambda^3 K$$

By following this process, we obtain

$$H(S^n(g_0), S^{n+1}(g_0)) \leq F\left(\frac{2a_1 + b_2}{1 - a_2}\right)^{n-2} K$$

As $F$ is strictly increasing and $\tau$ is greater than zero, we have

$$H(S^n(g_0), S^{n+1}(g_0)) < \left(\frac{2a_1 + b_2}{1 - a_2}\right)^{n-2} K$$
$$H(S^n(g_0), S^{n+1}(g_0)) \leq \lambda^{n-2} K$$

For $m > n$, we need to show $\{g_n\}$ is a Cauchy sequence in $G$. Using the triangular inequality, we obtain

$$H(S^n(g_0), S^m(g_0)) \leq H(S^n(g_0), S^{n+1}(g_0)) + H(S^{n+1}(g_0), S^{n+2}(g_0)) + \cdots +$$
$$H(S^{m-1}(g_0), S^m(g_0))$$
$$\leq \lambda^{n-2} K + \lambda^{n-1} + \cdots + \lambda^{m+n-3} K$$

$$\leq \left( \frac{\lambda^{n-1}}{1-\lambda} \right) K \quad \text{(a geometric series converging to zero).}$$

$$\text{Hence} \quad \lim_{n \to \infty} H(S^m(g_0), S^n(g_0)) = 0$$

This demonstrates that the sequence $\{g_n\}$ is Cauchy in $G$, implying the existence of an element $z \in G$ such that

$$\lim_{n \to \infty} g_n = \lim_{n \to \infty} S^n g_0 = z.$$

Now we have to prove that $z$ is a fixed point of $S$. As $S$ is orbitally continuous, so we get

$$z = \lim_{n \to \infty} S(S^n(g_0)) \in Sz$$

This shows that $z$ is a fixed point of $S$.

**Definition 2.12** Let $(G, d)$ be a complete metric space. A mapping $S : G \to CB(G)$ is a weak convex contraction if it satisfies

$$H(S^2(g), S^2(h))$$
$$\leq a_0 d(g, h) + b_0 d(h, S(g)) + a_1 d(S(g), (Sh)) + b_1 d(S(h), S^2(g))$$

for all $g, h \in G$ and $b_0, b_1 \geq 0$ and $0 < a_0, a_1 < 1$ and $a_0 + a_1 < 1$.

Here is an extention of Theorem 3 of Berinde and Berinde [2] for a weak convex contraction.

**Theorem 2.13** Let $(G, d)$ be a complete metric space and $S : G \to CB(G)$ a multivalued weak convex contraction. Then

(i) $Fix(S) \neq \phi$;

(ii) For every initial point $g_0 \in G$, there exists a sequence $\{g_n\}_{n=0}^{\infty}$ generated by the operator $S$ that converges to a fixed point $u$ of $S$, for which the following estimates hold:

$$d(g_n, u) \leq \frac{h^n}{1-h} d(g_0, g_1), n = 0, 1, 2, ..., \tag{12}$$

$$d(g_n, u) \leq \frac{h}{1-h} d(g_{n-1}, g_n), n = 0, 1, 2, ..., \tag{13}$$

for a certain real number $h < 1$.

**Proof:** (i) Let $q > 1$. Let $g_0 \in G$ and $g_1 \in Sg_0$. We consider two cases based on the Hausdorff distance between the iterates of $S$.

**Case 1:** If $H(S^2 g_0, S^2 g_1) = 0$, then $S^2 g_0 = S^2 g_1$.

Now $S^2(g_0) = S(S(g_0))$ implies that $S(g_0) = S(g_1)$ (cf. Remark 1.3(i)). Therefore $g_1 \in Sg_1$ gives $Fix(S) \neq \phi$ and the proof is complete.

**Case 2:** Let $H(S^2 g_0, S^2 g_1) \neq 0$. By Lemma 1.28, there exists $g_2 \in Sg_1$ such that $d(g_1, g_2) \leq qH(S^2 g_0, S^2 g_1)$.

$$\leq q[a_0 d(g_0, g_1) + b_0 d(g_1, Sg_0) + a_1 d(Sg_0, Sg_1) + b_1 d(Sg_1, S^2 g_0)]$$
$$\leq q[a_0 d(g_0, g_1) + b_0 d(g_1, g_1) + a_1 d(g_1, g_2) + b_1 d(g_2, g_2)]$$
$$\leq \frac{q}{(1-a_1)} a_0 d(g_0, g_1)$$

$$\leq q_1 a_0 d(g_0, g_1) \qquad where \quad q_1 = \frac{q}{(1 - a_1)}$$

We take $q_1 > 1$ such that

$$h = q_1 a_0 < 1.$$

Hence

$$d(g_1, g_2) < h d(g_0, g_1).$$

If $H(S^2 g_1, S^2 g_2) = 0$, then $S(S(g_1)) = S(S(g_2))$, i.e., $g_2 \in Sg_2$.
Assume that $H(S^2 g_1, S^2 g_2) \neq 0$. Again by Lemma 1.28, there exists $g_3 \in Sg_2$ such that

$$d(g_2, g_3) \leq h d(g_1, g_2).$$

In this way, we construct an orbit $\{g_n\}_{n=0}^{\infty}$ at $g_0$ for $S$ satisfying

$$d(g_n, g_{n+1}) \leq h d(g_{n-1}, g_n), \quad n = 1, 2, \ldots \tag{14}$$

By (14), we obtain inductively

$$d(g_n, g_{n+1}) \leq h^n d(g_0, g_1). \tag{15}$$

Hence

$$d(g_{n+k}, g_{n+k+1}) \leq h^{k+1} d(g_{n-1}, g_n), \quad k \in N, n \geq 1. \tag{16}$$

Similarly, by (15), we have

$$d(g_n, g_{n+p}) \leq \frac{h^n (1 - h^p)}{1 - h} d(g_0, g_1), n, p \in N, \tag{17}$$

Considering $0 < h < 1$, it can be concluded that the sequence $\{g_n\}_{n=0}^{\infty}$ constitutes a Cauchy sequence. Given that $(G, d)$ is a complete metric space, it follows that the sequence $\{g_n\}_{n=0}^{\infty}$ converges. Let

$$u = \lim_{n \to \infty} g_n. \tag{18}$$

Then by (*) in the proof of Lemma 1.28, we get

$$\begin{aligned}
d(u, Su) &\leq d(u, g_{n+1}) + d(g_{n+1}, Su) \\
&\leq d(u, g_{n+1}) + H(S^2 g_n, S^2 u) \\
&\leq d(u, g_{n+1}) + [a_0 d(g_n, u) + b_0 d(u, Sg_n) + \\
&\quad a_1 d(Sg_n, Su) + b_1 d(Su, S^2 g_n)]
\end{aligned} \tag{19}$$

Letting $n \to \infty$ in (19) and using the fact that $g_{n+1} \in Sg_n$ imply by (18) that $d(u, Sg_n) \to 0$, as $n \to \infty$. So we get

$$d(u, Su) = 0$$

As $Su$ is closed, so $u \in Su$.

(ii) Let $p \to \infty$ in (17). Then $h^p$ approaches 0 and so we get

$$d(g_n, u) \leq \frac{h^n}{1-h} d(g_0, g_1).$$

This proves (12).

$$d(g_{n+k}, g_{n+k+1}) \leq h^{k+1} d(g_{n-1}, g_n), \quad k \in N.$$

In the same way when $k \to \infty$ in (16), we get

$$d(g_n, u) \leq \frac{h^n}{1-h} d(g_0, g_1). \tag{20}$$

At the end of this section, we present relation among various concepts, used in this work in the form of a diagram:

**1- Contraction**
**2- Convex Contraction**
**3- Hardy and Rogers Convex Contraction**
**4- Weak Convex Contraction**
**5- Generalized Convex Contraction**
**6- F-Contraction**

## 2.3 Diagram

Here the arrow stands for the inclusion.

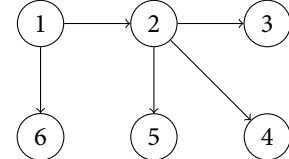

## 3 Application

Let $G = C[a, b]$ represent the vector space of all continuous real-valued functions on $[a, b]$ endowed with the usual metric. Then $(G, d)$ is a complete metric space.

The non-linear Fredholm integral equation is given as follows:

$$h(t) = v(t) + \frac{1}{b-a} \int_a^b k(t, s, h(s)) \, ds \tag{21}$$

where $t, s \in [a, b]$, $k : [a, b] \times [a, b] \times G \to \mathbb{R}$ and $v : [a, b] \to \mathbb{R}$ are continuous and $v(t)$ is a given function in $G$.

The solution of non-linear Fredhalm integral equation has been obtained for Hardy and Rogers convex contraction in [9]. We apply our Theorem 2.2 to solve non-linear Fredhalm integral equation for a Chatterjea convex contraction.

**Theorem 3.1:** Let $G = C[a, b]$ be the equipped with the usual metric.

Assume that (i) $S : G \to G$ is given by

$$Sg(t) = v(t) + \frac{1}{b - a} \int_a^b k(t, s, g(s)) \, ds \tag{22}$$

(ii) For $a_1, a_2, b_1, b_2 \in [0, 1)$, $g, h \in G$ with $g \neq h$ and $s, t \in [a, b]$, we have

$$|k(t, s, S(g)) - k(t, s, S(h))| \leq a_1 d(g(s), S(h(s))) + a_2 d(S(h(s)), S^2 h(s) \\ + b_1 d(h(s), S(g(s))) + b_2 d(S(g(s)), S^2 g(s)) \tag{23}$$

If $S$ is orbitally continuous, the integral operator defined by equation (22) possesses a unique solution $z \in G$. Moreover, for any initial value $g_0 \in G$, it holds that $S(g) \neq g_n$ for all $n \in \mathbb{N} \cup \{0\}$. Consequently, we have $\lim_{n \to \infty} S^n(g_0) = z$.

**Proof**

Let $g_0 \in G$ be an arbitrary point. Define a sequence $\{g_n\}$ in $G$ by $S(g_n) = S^{(n+1)}(g_0)$ for all $n \geq 0$. By (22), we have,

$$g_{n+1} = S(g_n(t)) = v(t) + \frac{1}{b - a} \int_a^b k(t, s, g_n(s)) \, ds$$

We must demonstrate that $S$ is a Chatterjea two sided convex contraction on $C[a, b]$. Use of (22) and (23) yields

$$|S^2 g(t) - S^2 h(t)|$$

$$= \frac{1}{|a - b|} \left| \int_a^b k(t, s, Sg(g)) ds - \int_a^b k(t, s, Sh(s)) ds \right|$$

$$\leq \frac{1}{b - a} \int_a^b \{|k(t, s, Sg(s)) - k(t, s, Sh(s))|\} \, ds$$

$$\leq \frac{1}{|b - a|} \int_a^b \{|(a_1|g(s) - Sh(s)| + a_2|Sh(s) - S^2 h(s)| + \\ b_1|h(s) - Sg(s)| + b_2|Sg(s) - S^2 g(s)|)|\} \, ds$$

$$\leq \frac{(a_1 + a_2 + b_1 + b_2)}{|b - a|} \int_a^b \max \{|g(s) - Sh(s)|, |Sh(s) - S^2 h(s)|, \\ |h(s) - Sg(s)|, |Sg(s) - S^2 g(s)|\} \, ds$$

$$\leq \frac{k}{b - a} \max \{|g(s) - Sh(s)|, |Sh(s) - S^2 h(s)|, \\ |h(s) - Sg(s)|, |Sg(s) - S^2 g(s)|\} \int_a^b ds$$

$$\leq k \max \{d(g, S(h)), d(S(h), S^2(h)), d(h, S(g)), d(S(g), S^2(g))\}$$

where $k = a_1 + a_2 + b_1 + b_2 < 1$ for all $g, h \in G$ with $g \neq h$.

Since $G$ is a complete metric space, therefore the iterative process converges to a specific point $z \in G$ (i.e. $\lim_{n \to \infty} g_n = z$). By orbital continuity of $S$, we can establish that $z$ is a fixed point of $S$. Thus all the conditions of Theorem 2.2 are satisfied and so by it's conclusion the non-linear Fredholm integral operator $S$ defined by (22) has a unique solution.

Now we provide an example to demonstrate the application of Theorem 3.1

**Example 3.2:**

Let us consider the operator $S : C[0, 1] \to C[0, 1]$ defined as:

$$Sg(t) = \frac{1}{2}g(t) + \int_0^1 \left( \frac{1}{3}g(s) + \frac{1}{4}(t + s) \right) ds, \tag{24}$$

where $g \in C[0, 1]$, and $t, s \in [0, 1]$. The kernel $k(t, s, g(s))$ is given by:

$$k(t, s, g(s)) = \frac{1}{3}g(s) + \frac{1}{4}(t + s).$$

Let us choose the following constants:

$$a_1 = \frac{1}{5}, \quad a_2 = \frac{1}{6}, \quad b_1 = \frac{1}{7}, \quad b_2 = \frac{1}{8},$$

which satisfy the requirement:

$$a_1 + a_2 + b_1 + b_2 = \frac{1}{5} + \frac{1}{6} + \frac{1}{7} + \frac{1}{8} < 1.$$

Let $g, h \in C[0, 1]$. We must demonstrate that:

$$|k(t, s, S(g)) - k(t, s, S(h))| \leq a_1|g(s) - S(h(s))| + a_2|S(h(s)) - S^2(h(s))| +$$
$$b_1|h(s) - S(g(s))| + b_2|S(g(s)) - S^2(g(s))|.$$

Substituting the kernel $k(t, s, g(s)) = \frac{1}{3}g(s) + \frac{1}{4}(t + s)$, we get:

$$|k(t, s, S(g)) - k(t, s, S(h))| = \left| \frac{1}{3}S(g(s)) - \frac{1}{3}S(h(s)) \right|.$$

Substituting the expressions for $S(g(s))$ and $S(h(s))$, we have:

$$\left| \frac{1}{3} \left( \frac{1}{2}g(s) + \int_0^1 k(t, u, g(u))du \right) - \frac{1}{3} \left( \frac{1}{2}h(s) + \int_0^1 k(t, u, h(u))du \right) \right|.$$

By the given nature of $g$ and $h$, the subsequent inequality holds:

$$|k(t, s, S(g)) - k(t, s, S(h))| \leq a_1|g(s) - S(h(s))| + a_2|S(h(s)) - S^2(h(s))| +$$
$$b_1|h(s) - S(g(s))| + b_2|S(g(s)) - S^2(g(s))|.$$

So $S$ satisfies (23). The operator $S$ being continuous is orbitally continuous.

Thus, all the requirements of Theorem 3.1 are satisfied and so the operator $S$ defined by (24) has a unique solution.

## 4 Conclusion

In this work, we have proved fixed point theorems for single-valued convex contraction mappings in b-metric spaces. Generalizing, some of these results for multivalued convex contraction mappings, an analogue of well-known theorems of Nadler and Istratescu are obtained. A result for an F-convex contraction is also established. A diagram is included here to provide an insight for the relationship among various convex contractions. A special case of Theorem 2.11 is applied to solve a non-linear Fredholm integral equation in the context of a Chatterjea two-sided convex contraction.

## 5 Open problems

Establish Theorems 2.2, 2.5, 2.10 and 2.13 for common fixed points and coincidences of convex contractions.

## Acknowledgments

The authors are grateful to the reviewers for their valuable remarks, which have helped us improve the presentation of this manuscript.

## Author contributions

**Conceptualization:** Abdul Rahim Khan.

**Funding acquisition:** Hamed H. Al-Sulami.

**Investigation:** Hamed H. Al-Sulami, Muhammad Rashid, Faiza Shabbir.

**Methodology:** Muhammad Rashid.

**Project administration:** Abdul Rahim Khan.

**Writing – original draft:** Abdul Rahim Khan, Muhammad Rashid, Faiza Shabbir.

**Writing – review & editing:** Hamed H. Al-Sulami.

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
