## [Decision Letter · Decision Letter 0]

5 Feb 2025

PONE-D-24-59912Fixed Points of Multivalued Convex Contractions with ApplicationPLOS ONE

Dear Dr. Khan,

Thank you for submitting your manuscript to PLOS ONE. After careful consideration, we feel that it has merit but does not fully meet PLOS ONE’s publication criteria as it currently stands. Therefore, we invite you to submit a revised version of the manuscript that addresses the points raised during the review process.

**ACADEMIC EDITOR: ** After reviewing the manuscript, I believe that it has significant potential, but some minor revisions are necessary before it can be considered for final acceptance. The revisions will enhance clarity and strengthen the overall quality of the work. Once these minor revisions are made, the manuscript will be in a strong position for publication. Please resubmit the revised version at your earliest convenience.

We look forward to receiving your revised manuscript.

Kind regards,

Rizwan Anjum

Academic Editor

PLOS ONE

Journal Requirements:

Reviewers' comments:

Reviewer's Responses to Questions

**Comments to the Author**

1. Is the manuscript technically sound, and do the data support the conclusions?

Reviewer #1: Yes

Reviewer #2: Yes

2. Has the statistical analysis been performed appropriately and rigorously? 

Reviewer #1: N/A

Reviewer #2: I Don't Know

3. Have the authors made all data underlying the findings in their manuscript fully available?

Reviewer #1: Yes

Reviewer #2: Yes

4. Is the manuscript presented in an intelligible fashion and written in standard English?

Reviewer #1: Yes

Reviewer #2: Yes

5. Review Comments to the Author

Reviewer #1: The report of the paper: Fixed Points of Multivalued Convex Contractions with Application

Dear Editor, PLOS ONE

This paper contains:

• Abstract (adequate);

• Introduction and Preliminaries (very adequate with clear Definitions,

Lemmas, Remarks and Examples);

• Main Results (6 new theorems);

• Application (1);

• Conclusion (sufficient)

• References (27 references)

I have read this paper and I can say that it is well-structured, clearly written,

and addresses an important area of study in fixed point theory. The theoretical

results are robust and are complemented by a practical application.

Hence, I strongly recommend it for publication in PLOS ONE after the

following minor revision:

1. In page 1, under Definition 1.1, add for any g, h ∈ G

2. Under pages 17 and 18, replace \varnothing with \emptyset.

3. On page 20, lines 11 and 13 from the bottom should be combined with

line 20 from the bottom.

4. Check the page nunmbers of references [9] and [26]

Reviewer #2: The authors have established fixed point results for single-valued convex contraction-type mappings in b-metric spaces and extended some of their results for multivalued convex contractions and F-convex contractions. In this manuscript, equivalents of the well-known theorems of Nadler and Istratescu are demonstrated for multivalued convex contractions. The new result established in Theorem 2.11 is applied to solve a non-linear Fredholm integral equation in the context of a Chatterjea two-sided convex contraction.

The results obtained in this article are of fundamental nature and their proof seems correct and logical. The paper is well written, and the use of terminology and symbols is consistent. All references are properly mentioned at appropriate places. Appropriate language is used throughout the manuscript. In my view this work lays the foundation for more investigation in this important area of mathematics. I strongly recommend the publication of this paper in PLOS ONE.

6. PLOS authors have the option to publish the peer review history of their article (what does this mean?). If published, this will include your full peer review and any attached files.

Reviewer #1: No

Reviewer #2: No

---

## [Author Response · Author response to Decision Letter 1]

3 Mar 2025

Rebuttal Letter

Manuscript ID: PONE-D-24-59912

Title: Fixed Points of Multivalued Convex Contractions with Application

Journal: PLOS ONE

Dear Editors,

We sincerely appreciate the reviewers' insightful comments and valuable suggestions. Below, we provide our point-by-point responses along with the corresponding revisions made to the manuscript.

Reviewer #1 Comments and Responses

1. Comment: On page 1, under Definition 1.1, add "for any g,h∈Gg,h \in Gg,h∈G" for clarity.

Response: We believe the reviewer intended to refer to Definition 1.2 rather than Definition 1.1. Accordingly, we have added "for any g,h∈Gg,h \in Gg,h∈G" in Definition 1.2 to enhance clarity.

2. Comment: On pages 17 and 18, replace ∅\varnothing∅ with ∅\emptyset∅.

Response: In the updated PLOS ONE template, the relevant section now appears on page 13. We have replaced all occurrences of ∅\varnothing∅ with ∅\emptyset∅ to maintain consistency in notation.

3. Comment: On page 20, lines 11 and 13 from the bottom should be combined with line 20 from the bottom.

Response: In the revised PLOS ONE template, this section now appears on page 14. We have merged the mentioned lines to improve readability and coherence.

4. Comment: Check the page numbers of references [9] and [26].

Response:

• Reference [9]: The issue notation has been updated to 6(1).

• Reference [26]: The missing volume number has been added.

Final Remarks

All suggested revisions have been carefully implemented. We sincerely appreciate the reviewers’ constructive feedback, which has contributed to improving the manuscript. We kindly request your consideration of our revised submission.

Best regards,

Abdul Rahim Khan

abdulrahimkhan@isp.edu.pk

---

## [Editor Report · Decision Letter 1]

12 Mar 2025

Fixed Points of Multivalued Convex Contractions with Application

PONE-D-24-59912R1

Dear Dr. Khan,

We’re pleased to inform you that your manuscript has been judged scientifically suitable for publication and will be formally accepted for publication once it meets all outstanding technical requirements.

Kind regards,

Rizwan Anjum

Academic Editor

PLOS ONE

Additional Editor Comments (optional):

All minor comments have been addressed by the authors, and the paper is now accepted for publication. This is a high-quality paper that is expected to attract significant interest from researchers working in this area.
---

## [Editor Report · Acceptance letter]

PONE-D-24-59912R1

PLOS ONE

Dear Dr. Khan,

I'm pleased to inform you that your manuscript has been deemed suitable for publication in PLOS ONE. Congratulations! Your manuscript is now being handed over to our production team.

Kind regards,

on behalf of

Dr. Rizwan Anjum

Academic Editor

PLOS ONE